# Role of *IL6R* Genetic Variants in Predicting Response to Tocilizumab in Patients with Rheumatoid Arthritis

**DOI:** 10.3390/pharmaceutics14091942

**Published:** 2022-09-14

**Authors:** Luis Sainz, Pau Riera, Patricia Moya, Sara Bernal, Jordi Casademont, Cesar Díaz-Torné, Ana Milena Millán, Hye Sang Park, Adriana Lasa, Héctor Corominas

**Affiliations:** 1Rheumatology Department, Hospital de la Santa Creu i Sant Pau, 08041 Barcelona, Spain; 2Department of Medicine, Universitat Autònoma de Barcelona (UAB), 08193 Barcelona, Spain; 3Institut d’Investigació Biomèdica Sant Pau (IIB SANT PAU), 08041 Barcelona, Spain; 4Pharmacy Department, Hospital de la Santa Creu i Sant Pau, 08025 Barcelona, Spain; 5CIBER de Enfermedades Raras (CIBERER), Instituto de Salud Carlos III, 28029 Madrid, Spain; 6Genetics Department, Hospital de la Santa Creu i Sant Pau, 08025 Barcelona, Spain; 7Internal Medicine Department, Hospital de la Santa Creu i Sant Pau, 08025 Barcelona, Spain

**Keywords:** *IL6R*, genetic variants, tocilizumab, rheumatoid arthritis, predictive factors

## Abstract

Rheumatoid arthritis (RA) is a prevalent autoimmune disease characterized by chronic arthritis that may lead to irreversible joint damage and significant disability. Patients with RA are commonly treated with Tocilizumab (TCZ), an IL-6 receptor (IL-6R) antagonist, but many patients refractorily respond to this therapy. Identifying genetic biomarkers as predictors of TCZ response could be a key to providing a personalized medicine strategy. We aimed to evaluate whether functional single nucleotide polymorphisms (SNPs) in the *IL6R* gene could predict TCZ response in patients with RA. We retrospectively included 88 RA patients treated with TCZ. Six SNPs previously described in the *IL6R* gene (rs12083537, rs11265618, rs4329505, rs2228145, rs4537545, and rs4845625) were genotyped in DNA samples from these patients. Using parametric tests, we evaluated the association between these polymorphisms and clinicopathological features. Responses to treatments were assessed at six months using three variables: a quantitative improvement in Disease activity score including 28 joints (DAS28), a satisfactory European League Against Rheumatism (EULAR) response, and low disease activity (LDA) achievement. The three response variables studied were associated with genetic variant rs4845625, and no association was found with the other five SNPs. Our findings support the potential clinical value of SNPs in the *IL6R* gene as predictive biomarkers for TCZ response.

## 1. Introduction

Rheumatoid arthritis (RA) is a common autoimmune and inflammatory disease with an estimated worldwide prevalence ranging from 0.2 to 1% [1]. The most recent estimate in adults in Spain, published in 2016, reported a prevalence of 0.82% (95% CI, 0.59–1.15) [2]. RA etiology is complex as it involves the interplay of environmental triggers and genotype, factors known to play an important role in RA physiopathology. The most relevant genes associated with RA are involved in immunity activation and regulation, antigenic presentation, and proinflammatory cytokines [3]. RA is characterized by synovial inflammation, hyperplasia, and the progressive destruction of bones and cartilages in multiple joints [3]. If not treated correctly, these characteristics cause progressive disability, systemic complications, and reduced life expectancy [4]. Over the last 20 years, treatment has changed substantially thanks to the development of targeted biologic and non-biologic disease-modifying anti-rheumatic drugs. These high-efficacy therapies enabled the improved control of the disease, and their diverse mechanisms of action, targeting various immune pathways, broadened the treatment armamentarium. Therapeutic strategies have also changed. Focus is now centered on early treatments and a treat-to-target approach where the main therapeutic goal is to achieve remissions or low disease activity (LDA) in order to prevent structural damage and maintain a quality of life [5,6,7,8,9,10].

The choice of treatments for RA is currently a trial-and-error process as no consistent clinical, biochemical, or genetic biomarkers that predict response to biological disease-modifying anti-rheumatic drug (bDMARD) have yet been identified. As delays in finding effective treatment can lead to disease progression and a poorer quality of life, establishing biomarkers of different responses is of paramount importance [11,12,13]. Because genetics can explain much of the inter-individual response to treatments, the study of single nucleotide polymorphisms (SNPs) has gained increasing interest in recent years. In the field of pharmacogenomics, such research has focused on genetic biomarkers relative to MTX [14,15,16,17] and anti-tumor necrosis factor (anti-TNF) treatments [18,19,20,21,22,23,24].

Tocilizumab (TCZ) is a first-line bDMARD that competitively inhibits the binding of interleukin-6 (IL-6) to its soluble or membrane receptor (IL-6R). It is effective in the treatment of patients with moderate to severe RA and European League Against Rheumatism (EULAR) response rates of between 62 and 80 [13,25,26,27]. Nevertheless, due to pharmacoeconomic considerations, it is generally only prescribed after an inadequate response to methotrexate (MTX) or anti-TNF agents ([28,29,30]). Despite the effectiveness of TCZ in RA [13], some patients do not respond adequately to this drug. Taking into account that TCZ blocks the action of IL-6R, we hypothesized that functional variations in this gene could affect TCZ effectiveness. Previous studies assessed the influence of *IL6R* genetic variants on TCZ responses. However, the results hitherto published are inconsistent and produce contradictory results, and they are, therefore, not applicable to clinical practice [31,32,33,34,35]. The aim of the present study was to evaluate whether genetic variants in the *IL6R* gene are associated with responses to TCZ in patients with RA. 

## 2. Materials and Methods

### 2.1. Study Population

We conducted a retrospective cohort study that included 88 RA patients treated with TCZ between 2016 and 2021. The following sociodemographic and clinical data were collected from electronic medical records: age, gender, age at diagnosis, previous treatments, comorbidities, TCZ starting date, TCZ dose, administration route, concomitant treatments, C-reactive protein (CRP), rheumatoid factor (RF), anti-citrullinated protein antibody (ACPA), DAS28 at treatment initiation and at 6 months, and EULAR response at 6 months. DAS28 is a simplified disease activity score frequently used in trials and clinical practice that includes counting tender and swollen joints, the general health state evaluated by the patient, and acute phase reactants in blood tests. Response variables were determined at the baseline and at 6 months of TCZ treatment. Responses to treatment were assessed using three variables: a quantitative improvement in DAS28, a satisfactory EULAR response, and low disease activity (LDA). According to the EULAR guidelines, we considered a EULAR response to be satisfactory when the DAS28 improvement was greater than 1.2 and the presented DAS28 was lower than 3.2. We considered LDA when patients reached a DAS28 lower than 3.2 at 6 months [36,37].

The study was approved by the institutional ethics committees and registered at clinicaltrials.gov (protocol code: IIBSP-IIL-2020-148). All participants provided written informed consent for blood sampling and genetic analyses.

### 2.2. Genetic Studies

SNPs were selected according to published data concerning their functional relevance in the IL6 receptor. Those included were rs12083537, rs11265618, rs4329505, rs2228145, rs4537545 and rs4845625 (Table 1). The rationale for the selection of these SNPs was detailed in previous studies [31,32,33,34,38]. All SNPs presented a minor allele frequency (MAF) over 0.10 in European population according to the ALFA project [39]. 

Genomic DNA was automatically extracted from peripheral whole-blood samples (Autopure, Qiagen, Hilden, Germany). SNPs were analyzed by real-time PCR using TaqMan SNP genotyping assays (Applied Biosystems, Foster City, CA, USA). The IDs for each Taqman SNP assays were the following: rs12083537 (C_30997483_10), rs11265618 (C_30997439_10), rs4329505 (C_26292281_20), rs2228145 (C_16170664_10), and rs4537545 (C_11258666_20). A custom Taqman assay was used for rs4845625. All cases were successfully genotyped. 

#### Statistical Analyses

The Hardy–Weinberg equilibrium was assessed for each analyzed SNP using a chi-square test. Codominant, dominant, and recessive models of inheritance were considered to assess associations with outcome variables whenever appropriate.

Quantitative data were expressed as the mean (SD) for normally distributed variables. Normality was assessed using the Shapiro–Wilk test. The Student’s t-test or ANOVA was applied for normally distributed variables depending on the number of groups compared. We analysed the bivariate association for qualitative dichotomous variables using Pearson’s χ^2^ or Fisher’s exact test. Associations between the various SNPs and the qualitative response variables were tested using χ^2^ tests. 

All tests were two-sided and a significance level of 5% (α = 0.05) was considered significant. All statistical analyses were performed using IBM-SPSS (version 26.0, IBM, Armonk, NY, USA).

## 3. Results

### 3.1. Characteristics of the Patients

We studied 88 patients with RA diagnosed at a mean age of 46.6 years (SD 15.86). Mean disease duration was 16.5 (SD 11.9) years. Most patients were female (86.4%). Table 2 summarizes baseline demographic and clinical characteristics.

The mean numbers of conventional cDMARD and bDMARD treatments received prior to TCZ were 2.39 (SD 1.3) and 1.48 (SD 1.4), respectively. 

Baseline disease activity, measured by DAS28, was high, with a mean of 5.4 (SD 0.9). We were unable to include 4 of the 88 patients (4.5%) because the treatment was discontinued prematurely due to a loss of follow-up in 3 patients and the observation of a hypersensitivity reaction in 1 patient. Thus, these patients were not included in the efficacy analysis. After 6 months, the mean decrease in DAS28 was 2.9 (SD 1.3) and the satisfactory EULAR response rate was 73.9%. 

None of the response variables studied showed statistically significant differences according to sex, tobacco exposure, seropositivity, the number of previous cDMARD or bDMARD, age at diagnosis, and body mass index (BMI). However, we observed two trends concerning these variables: EULAR satisfactory responses were higher in women than in men (79.7% vs. 60%, *p* = 0.16) and non-smokers showed a trend towards greater DAS28 improvements than smokers (3.04 vs. 2.35, *p* = 0.08).

### 3.2. Genetic Determinants and Response to Treatment

The genotypic frequencies of the six SNPs studied were in Hardy–Weinberg equilibrium. As expected from the allele frequency population-based studies, we found that the mutated allele (C) of the SNP rs4845625 was more frequent than the wild-type (T) [39]. 

In the univariate analysis, only one SNP, rs4845625, showed a statistically significant association with the response outcomes studied (Table 3). When considering DAS28 improvement, six months after initiating TCZ treatment, we observed differences between the three genotypes compatible with an additive genetic model. The TT carriers for rs4845625 had a greater decrease (3.62) in DAS28 than those with the CT genotype (3.06) and the CC genotype (2.36) (*p* = 0.015).

Patients carrying the T allele (TT + CT) also reached more satisfactory EULAR response rates than CC carriers (87.3% vs. 58.6%, *p* = 0.03). Accordingly, when assessing LDA rates, the TT and CT genotypes also demonstrated better outcomes (86.5% vs. 56.6%, *p* = 0.04. OR (95% CI) = 4.95 (1.5–15.56)).

We did not perform a multivariate analysis as there were no statistically significant associations that could act as confounding variables. 

As for the other five SNPs studied, no statistically significant associations were found (Table 3). The results showed a trend towards an association with response variables for rs4329505 and rs11265618, although they did not reach statistical significance. Patients carrying the CC genotype for rs4329505 seemed to have a poorer reduction in DAS28 at 6 months than those with the TT + CT genotypes (1.76 vs. 2.95, *p* = 0.06). For the rs11265618 variant, the CC genotype (dominant homozygous) appeared to have superior satisfactory EULAR response rates than CT + TT genotypes (82.5% vs. 66.7%, *p* = 0.106).

## 4. Discussion

In this study, we assessed the associations between six SNPs in the *IL6R* gene and the response outcomes at 6 months in patients with RA. We found an association between rs4845625 and the three response variables used: DAS28 improvement, EULAR response, and LDA achievement.

As far as we know, this is the first study to show that rs4845625 could be a potential biomarker of response to TCZ in RA patients. Our results suggest that patients carrying TT and TC genotypes present higher satisfactory EULAR response rates when treated with tocilizumab (85.7–92.3% compared to the 73.9% of the whole cohort). However, larger prospective studies are needed to validate our results and for implementing rs4845625 genotyping in clinical practice. 

This SNP is located at Chr1 (q21.3) g.154422067 in intron 7 of the *IL6R* gene. In silico tools such as the ESE Finder predict that this SNP could alter the splicing process and, therefore, modify the final amount of a functional protein. However, further analyses are required to confirm this hypothesis. 

Current evidence shows several associations between rs4845625 and inflammatory pathways. Shah T et al. reported that rs4845625 is associated with plasma concentrations of IL6. However, the contribution of the SNPs evaluated in plasma IL6 levels in their study was mild [40]. Associations of the allele rs4845625*T with increased levels of RCP, LDL, and apolipoprotein B were also found in a healthy cohort of 995 participants [38]. These results contrast with our findings of improved response outcomes in T allele carriers. A possible explanation for this discrepancy could be that a patient with a genetically determined pro-inflammatory basal state would have a greater chance to respond to treatments blocking these pathways. However, studies are lacking regarding associations between this genotype and inflammatory markers in the context of autoimmune diseases. Rs4845625 genetic variants have been linked to cardiovascular conditions in several reports, suggesting an important role of the IL-6 pathway ([41,42,43,44]), but we did not find previous research regarding rs4845625 and RA. 

Although the rs11265618 polymorphism in our study showed no statistically significant association with the response to treatment variables, we observed a trend towards a greater response of the homozygous genotype (CC). In 2018, Maldonado-Montoro et al. described an association within the CC genotype that showed a better response in terms of LDA after 12 months of TCZ therapy (OR (95% CI) = 12.11 (2.2–67.8)), but they were unable to demonstrate this regarding the EULAR response or the DAS28 improvement [31]. While our findings did not confirm previously reported results, we propose that further investigations and larger samples are needed to determine the potential usefulness of rs11265618 as a biomarker for TCZ therapy. 

Our results regarding rs4329505 also merit discussion. Despite the lack of statistical significance, our results using an additive genetic model suggest that the T allele could be associated with DAS28 improvements. Similar results have been reported previously. In a retrospective study including 79 RA patients, the CC genotype of rs4329505 was linked to a poorer response regarding less joint swelling at 3 months [33]. Supporting the biologic role of these SNP variants with inflammatory physiopathological pathways in autoimmune diseases, the CC genotype has shown an association with higher RCP levels in female diabetic patients. Interestingly, these changes were independent of IL-6 serum levels and were not observed in healthy individuals [45]. However, the study by Maldonado-Montoro et al. and the findings reported by Luxembourger et al. did not suggest that rs4329505 could predict responses to TCZ [31,34]. Overall, data supporting the role of rs4329505 genetic polymorphisms in the response to treatment with TCZ in RA patients are scarce and inconsistent, even though our findings seem to indicate a trend worth investigating in future studies. 

We did not find any associations between rs12083537, rs2228145, and rs4537545 genetic variants and the response to treatment outcomes. Diverse and conflicting results have been reported concerning the rs12083537 AA genotype. This variant was initially associated with a worse response when considering swollen joint count individually and when analyzed as a haplotype along with rs2228145 and rs4329505 variants [33]. In contrast, Maldonado-Montoro et al. and Luxembourger et al. reported improved responses to TCZ in the rs12083537 AA genotype when analyzing LDA at 12 months and EULAR response rates at 6 months, respectively [31,34]. These conflicting data contrast with the absence of associations in our study. Apart from the study by Enevold et al., there were no further reports of associations between rs2228145 and the response to TCZ in RA [33]. In contrast, possible physiopathological mechanisms of rs2228145 have been reproduced, suggesting a regulation of IL-6 bioactivity via the downregulation of cellular IL-6R [46,47]. 

Among the published studies about TCZ pharmacogenetics, we found only one study that followed a genome-wide association studies (GWAS) approach, which could be useful for avoiding missing potential biomarkers. However, none of the eight loci encountered clearly seemed to be involved in the physiopathology of RA or IL-6 pathways [35]. The inconsistent findings between studies and the lack of a robust biological basis on the most frequently reported SNP highlight the complexity of immune pathways in RA and the factors that modulate responses to TCZ.

The limitations of this study should be considered when interpreting results. First, the retrospective design and the relatively small sample size may have compromised potential associations between SNPs and response to TCZ. In particular, rs11265618 and rs4329505 homozygosity for the mutated allele is expected to be low, as mutated allele frequencies are 0.03 and 0.04. Nonetheless, the homogeneity of our cohort supports the validity of our findings. Second, as the present study was designed to study promising *IL6R* SNPs, we may not have detected other genetic variants as potential predictors to TCZ treatments. 

Further research in the pharmacogenomics of RA treatment is needed to establish robust evidence with clinical relevance that could lead to personalized medicine. In addition, further studies are needed to investigate the potential use of rs4845625 as a treatment biomarker of TCZ. 

## 5. Conclusions

Our results suggest that the *IL6R* polymorphism, rs4845625, is associated with DAS28 improvements, EULAR response rates, and an LDA at 6 months after initiating TCZ in RA patients. 

## Figures and Tables

**Table 1 pharmaceutics-14-01942-t001:** Selected functional polymorphisms in the *IL6R* gene.

refSeg	Genomic Position (GRCh38)	MAF	Alleles
rs12083537	chr1:154408627	0.21	A > **G**
rs11265618	chr1:154457616	0.17	C > **T**
rs4329505	chr1:154459944	0.17	T > **C**
rs2228145	chr1:154454494	0.40	A > **C**
rs4537545	chr1:154446403	0.41	C > **T**
rs4845625	chr1:154449591	0.43	**T** > C

Abbreviations: *IL6R*, Interleukin 6 Receptor; MAF, minor allele frequency reported in the European population. The minor allele is highlighted in bold.

**Table 2 pharmaceutics-14-01942-t002:** Baseline patient characteristics (*n* = 88).

Variables	*n* (%)	Mean (SD)
Female/Male	76 (86.4)/12 (13.6)	
Age at diagnosis (years)		46.57 (15.86)
Disease duration (years)		16.53 (11.87)
Erosive RA	47 (53.4)	
RF positive	58 (65.9)	
ACPA positive	57 (64.8)	
Smoking habit Non-smoker Ex-smoker Smoker	61 (69.3) 14 (15.9) 13 (14.8)	
Body mass index		28.64 (6.07)
Number of previous cDMARD		2.39 (1.32)
Number of previous bDMARD		1.48 (1.41)
TCZ administration (intravenous)	47 (53.4)	
Baseline DAS28-CRP		5.39 (0.98)

ACPA, anticitrullinated protein antibodies; CRP, C-reactive protein; DAS28-CRP, disease activity score including 28 joints (based on CRP); bDMARD, biological disease-modifying antirheumatic drug; ESR, erythrocyte sedimentation rate; RF, rheumatoid factor; SD, standard deviation; TCZ, tocilizumab.

**Table 3 pharmaceutics-14-01942-t003:** Associations between the considered SNPs and response outcomes at 6 months.

SNPs	Genotype (*n*)	DAS28 Improvement			Satisfactory EULAR Response Rates				Low Disease Activity Rates			
		Mean Absolute Value	Genetic Model	*p*	%	Genetic Model	OR (95% CI)	*p*	%	Genetic Model	OR (95% CI)	*p*
rs4845625	T/T (13)	3.62	Cod	**0.015**	92.3	Cod	-	**0.01**	90.9	Cod	-	**0.015**
	T/C (42)	3.07	Rec	**0.009**	85.7	Rec	4.83 (1.63–14.28)	**0.03**	85.4	Rec	4.95 (1.57 −15.55)	**0.004**
	C/C (29)	2.36			58.6				56.5			
rs11265618	C/C (57)	3.06	Cod	0.16	82.5	Cod	-	0.271	81.1	Cod	-	0.474
	C/T (24)	2.66	Dom	0.115	66.7	Dom	2.35 (0.82–6.71)	0.106	68.4	Dom	2.01 (0.65–6.22)	0.223
	T/T (3)	1.76			66.7				66.7			
rs4329505	T/T (58)	3	Cod	0.106	82.8	Cod	-	0.212	81.5	Cod	-	0.388
	T/C (23)	2.56	Dom	0.06	65.2	Dom	2.54 (0.88–7.29)	0.078	66.7	Dom	2.2 (0.7–6.8)	0.169
	C/C (3)	1.76			66.7				66.7			
	A/A (55)	2.94	Cod	0.717	78.2	Cod	-	0.335	81.6	Cod	-	0.454
rs12083537	A/G (24)	2.76	Dom	0.749	70.8	Dom	0.88 (0.3–2.54)	0.809	68.2	Dom	1.98 (0.66–5.95)	0.222
	G/G (5)	3.38			100.0				75.0			
	A/A (25)	3.01	Cod	0.483	80.0	Cod	-	0.888	78.3	Cod	-	0.841
rs2228145	A/C (44)	2.99	Dom	0.651	77.3	Dom	0.8 (0.26–2.54)	0.709	78.9	Dom	1.08 (0.33–3.52)	0.898
	C/C (15)	2.55			73.3				71.4			
	C/C (24)	2.98	Cod	0.702	83.3	Cod	-	0.666	81.8	Cod	-	0.690
rs4537545	C/T (46)	2.97	Dom	0.759	76.1	Dom	0.6 (0.18–2.04)	0.410	77.5	Dom	1.46 (0.42–5.1)	0.550
	T/T (14)	2.64			77.4				69.2			

Cod: Codominant; DAS28: disease activity score including 28 joints; Dom: dominant; OR: odds ratio; *p*: *p*-value; Rec: recessive; SNPs: single nucleotide polymorphisms. Significant values are highlighted in bold.

## Data Availability

The data presented in this study are available upon request from the corresponding author. The data are not publicly available due to the inclusion of clinical and personal information.

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
