# Peer review of "Role of IL6R Genetic Variants in Predicting Response to Tocilizumab in Patients with Rheumatoid Arthritis"

_pharmaceutics, 2022, doi:10.3390/pharmaceutics14091942_

Round 1

Reviewer 1 Report

This retrospective study assesses associations between 6 functional SNPs and response to IV Tocilizumab in 88 patients with RA at 6 months using DAS28, EULAR response and LDA as metrics of response.

The following questions and comments arise while reading this study:

1) The clinical relevance of the results comes down to the question: how much better would be the prediction of the response if you used the results from this study or how many patients would you reclassify as responders/ non-responders? The rates of response the authors have cited from the literature are 62-80%, so majority would be responders; in the study 73.9% were responders. How the results of this study help in improving this knowledge/ for clinical decision making?  The effect size from individual SNPs is usually very small.

2) Table 2: causes of discontinuation such as primary failure, secondary failure, adverse events listed. When were these outcomes observed? If within 6 months of the study then how does it affect interpretation of the study results and were these patients who censored accounted in the statistical analyses? Duration of treatment 45.91 - what units are used here?

3) page 4 - statistically insignificant comparisons should be mentioned even more cautiously if at all.

4) Table 3 - only 3 SNPs are included. What about the remaining 3 SNPs? Did authors account for multiple comparisons? P=0.05 is perhaps too permissive for genetic analyses.

5) In the introduction: results from previous studies on this subject ..."are inconsistent and not applicable to clinical practice". Why are they not applicable and how this study addresses this aspect?

Reviewer 2 Report

In this article Luis Sainz et al. investigated the role of polymorphic variants of IL6R gene as a possibly predicting response factor to tocilizumab in patients with rheumatoid arthritis. The study idea is general interesting – we need to try to finding the genetic variants which can be useful in choosing the best way of treatment. I have a few of comments:

1. In the Materials and Methods section, the authors should enter the assays numbers which they used for the study. Furthermore, they did not enter the name of the device on which the markings were made.

2. In the Study population section , the authors should explain what DAS28 means and briefly describe it.

3. In Table 2, the authors provide information that applies only to female, but the text states that men were also in the treatment group. Why are there no clinical data available for men when they entered the treatment group and were also tested for the presence of individual polymorphic variants? This needs to be corrected.

4. If authors describe research results to which they do not attach tables, they should indicate that these data were not included in the work. This information facilitates the analysis of the work and its results.

5. In discussion section, in the 191 line the authors write about the variant rs4845625 related to the risk of coronary artery disease. I don’t understand why the writers placed this information? What does this have to do with RA? This needs to be corrected.

Round 2

Reviewer 1 Report

The authors provided responses but not all these points were reflected in the revision: e.g. response to Q 1 (if no numeric estimate for % of patients reclassified, then at least this can be included in the discussion); Q 2: make sure that the 4 patients who were excluded are mentioned in the manuscript with the reasons for exclusion; Q4: Table 3 should include all SNPs that were studied; Q 5: was this clarification added to the introduction? I don't see any highlighted changes there.
